# CERTIFIED NEURAL APPROXIMATIONS OF NONLINEAR DYNAMICS

## ABSTRACT

Neural networks hold great potential to act as approximate models of nonlinear dynamical systems, with the resulting neural approximations enabling verification and control of such systems. However, in safety-critical contexts, the use of neural approximations requires formal bounds on their closeness to the underlying system. To address this fundamental challenge, we propose a novel, adaptive, and parallelizable verification method based on certified first-order models. Our approach provides formal error bounds on the neural approximations of dynamical systems, allowing them to be safely employed as surrogates by interpreting the error bound as bounded disturbances acting on the approximated dynamics. We demonstrate the effectiveness and scalability of our method on a range of established benchmarks from the literature, showing that it significantly outperforms the state-of-the-art. Furthermore, we show that our framework can successfully address additional scenarios previously intractable for existing methods– neural network compression and an autoencoder-based deep learning architecture for learning Koopman operators for the purpose of trajectory prediction.

## 1 INTRODUCTION

Nonlinear dynamical models are ubiquitous across science and engineering and play a central role in describing and designing complex cyber-physical systems (Alur, 2015). However, to verify and control such systems, it is often necessary to construct an abstraction: an approximation that locally simplifies the model or relaxes its nonlinearities (Derler et al., 2012; Khalil, 2002; Sastry, 1999). In general, an abstraction translates the *concrete model*—the system under study—into a simpler *abstract model* that is more amenable to analysis (Baier and Katoen, 2008; Clarke et al., 2018). A common method for synthesizing abstractions is known as *hybridization* and involves partitioning the state space into regions, each representing a state in a finite-state machine (Althoff et al., 2008; Asarin et al., 2007; Bak et al., 2016; Dang et al., 2010; Frehse, 2005; García Soto and Prabhakar, 2020; Henzinger and Wong-Toi, 1995; Li et al., 2020; Majumdar and Zamani, 2012; Prabhakar et al., 2015; Roohi et al., 2016). Neural networks with ReLU activations have emerged as a particularly effective approach to *hybridization*, as each network configuration implicitly induces a partition of the input domain into convex polytopes (Abate et al., 2022; Goujon et al., 2024; Villani and Schoots, 2023). This enables simultaneous learning of both the partitioning and the simplified dynamics.

For neural abstractions to be practically useful, properties inferred from the abstraction—such as those related to reachability or safety—must reliably transfer to the original system. Simulation-based techniques fall short, as they are non-exhaustive and may miss unsafe behaviour. Formal verification provides a sound alternative by exhaustively analyzing all possible inputs and outputs. Prior work has used SMT (Satisfiability Modulo Theories) solvers for this task (Fränzle et al., 2007; Abate et al., 2022; Solanki et al., 2025). SMT extends the Boolean Satisfiability Problem (SAT) to more complex formulas such as those involving linear real arithmetic or integers. This makes SMT solvers an indispensable tool for a range of applications; however, for neural network verification, the computational cost of SMT solvers severely limits the scale and expressivity of verifiable networks.

To overcome these limitations, we introduce a scalable framework for verification of neural abstractions that avoids reliance on expensive SMT solvers. Our method constructs certified first-order Taylor models to tightly bound a nonlinear system's behaviour, which enables us to employ neural network verification tools based on linear bound propagation to certify that the network's output remains

sufficiently close to the linearized dynamics. The use of Taylor models has achieved considerable success in reachability analysis for hybrid systems, particularly in the context of neural network-controlled systems (Huang et al., 2019; Ivanov et al., 2021; Huang et al., 2022). However, while these previous works have considered *local* reachability problems, we address a *global* closeness problem to obtain the same formal guarantees as prior work employing SMT solvers. To this end, we introduce a parallelizable partitioning and refinement scheme – achieving substantial verification speedups without loss of formal guarantees. Our approach allows us to handle higher-dimensional systems (up to 7D), and considerably larger neural networks than state-of-the-art methods. In summary, our approach addresses the primary bottleneck in neural abstractions Abate et al. (2022), namely the scalability of the $\epsilon$-closeness verification.

To exploit these new capabilities to handle higher-dimensional systems and larger networks, we extend neural abstractions to learning Koopman operators (Brunton et al., 2016; 2022) (Section 4.2). This application, which enables trajectory-level reasoning by representing nonlinear dynamics as linear operators in a high-dimensional space, previously posed a significant verification challenge, as the network's output represents predicted trajectory points (in our example, involving a final layer with over 100 states). Finally, to demonstrate the versatility of our framework, particularly for more general function approximation tasks, we extend neural abstractions beyond their native developments for dynamical systems, demonstrating their effective application to neural network compression (Section 4.3).

Our contributions are summarised as follows:

1. We introduce a novel method based on certified linearizations to eliminate the reliance on SMT solvers capable of handling nonlinear real arithmetic, thereby removing the primary computational bottleneck in prior approaches.

2. We introduce a parallelizable refinement strategy that enables adaptive verification of neural abstractions with nonuniform accuracy across the input domain.

3. We demonstrate that our approach outperforms existing methods on a variety of benchmarks.

4. We demonstrate the effectiveness of our approach on two novel applications that are beyond the capability of state-of-the-art methods: neural network compression and the discovery of Koopman operators.

We begin in Section 2 by formally introducing neural abstractions, followed by presenting our approach to certification in Section 3.

## 2 NEURAL APPROXIMATIONS OF NONLINEAR DYNAMICS

We begin by introducing the system dynamics, for which we will synthesise neural network abstractions over a bounded domain. Let $\mathcal{X} \subset \mathbb{R}^n$ denote the bounded input domain of interest, and suppose $f : \mathcal{X} \to \mathbb{R}^m$ is a continuous (nonlinear) function describing the system's dynamics. We focus on two classes of systems:

- **Continuous-time nonlinear systems**, described by differential equations of the form $\frac{dx}{dt} = f(x), \quad x \in \mathcal{X}$;

- **Discrete-time nonlinear systems**, described by difference equations of the form $x_{k+1} = f(x_k), \quad x_k \in \mathcal{X}$.

Given a dynamical system as described above, a neural abstraction is an $\epsilon$-close approximation of the dynamical system, as formally defined in Definition 1 below, which is a generalization of the definition of neural abstractions introduced in Abate et al. (2022) for continuous-time dynamical systems.

**Definition 1** (Neural Abstraction). *Consider a dynamical system described by function $f : \mathbb{R}^n \to \mathbb{R}^m$ and let $\mathcal{X} \subset \mathbb{R}^n$ be a region of interest. A feed-forward neural network $\mathrm{N} : \mathbb{R}^n \to \mathbb{R}^m$ defines a neural abstraction, also called a neural approximation, of $f$ with error bound $\epsilon > 0$ over $\mathcal{X}$, if it holds that $\forall x \in \mathcal{X} : \|f(x) - \mathrm{N}(x)\| \le \epsilon$ where $\|\cdot\|$ is the $L_\infty$-norm[1].*

---

[1]For the remainder of the paper, unless otherwise specified, all norms are $L_\infty$.

According to Definition 1, a neural abstraction of a dynamical system describes a dynamical system with a bounded additive disturbance $d$, such that any trajectory (solution to the differential equation) of the original dynamical system $f$ is also a trajectory of the perturbed system. Specifically, in the case where $f$ describes a continuous-time system, we have the following equation for the perturbed system:

$$\frac{dx}{dt} = N(x) + d, \quad \|d\| \leq \epsilon, \quad x \in \mathcal{X}, \tag{1}$$

where $\epsilon$ represents the maximal deviation between the neural network approximation $N(x)$ and the original function $f(x)$. In the discrete-time case, where $f$ defines an update rule $x^{k+1} = f(x^k)$, any trajectory (solution to the difference equation) of the original system $f$ is also a trajectory of the following discrete-time system:

$$x^{k+1} = N(x^k) + d, \quad \|d\| \leq \epsilon, \quad x \in \mathcal{X}. \tag{2}$$

As the disturbance $d$ is bounded by $\epsilon$, the original system response, defined by $f$, is always contained within that of the abstraction, defined by $N$ with disturbance $d$. Thus, the abstraction is sound, which enables formal guarantees to transfer from the abstraction to the concrete model.

**Remark.** *The applicability of our framework is not contingent on access to $f$ not its derivatives, but only on access to linear bounding functions, as we will show in Section 3. Linear bounding functions can be constructed in various ways with varying degrees of conservatism depending on the available information about $f$. We describe three such methods in Appendix D.*

### 2.1 TRAINING

To obtain a neural network approximation of a dynamical system, we train a neural network $N$ to minimize both the mean and maximum approximation error over a batch of sampled inputs $\{x_1, \ldots, x_M\}$. To this end, we assume for the remainder of the paper that the function $f$ has bounded output and define the following loss function:

$$\mathcal{L} = \frac{1}{M} \sum_{l=1}^{M} \|f(x_l) - N(x_l)\|_2 + \lambda_{\max} \max_{l \in \{1, \ldots, M\}} \|f(x_l) - N(x_l)\|_\infty, \tag{3}$$

where the parameter $\lambda_{\max} = 0.001$ balances the trade-off between minimizing the average error and controlling the worst-case error across the sampled domain. We focus on training neural abstractions with ReLU and LeakyReLU activation functions, although our approach is compatible with more general activation functions, as permitted by the underlying solver (neural network verification tool). For our approach to be sound while also scalable, we utilise a complete solver, specifically Marabou 2.0 (Wu et al., 2024). Further details regarding the network architecture, training procedure, and hyperparameters are provided in Section 4 and Appendix B. Once a neural network $N$ has been trained, the central challenge lies in certifying the accuracy, *i.e.*, formally establishing the relation $\|f(x) - N(x)\| \leq \epsilon$ between the neural network abstraction and the concrete model. In what follows, we present our primary contribution, a scalable verification approach for this problem based on an adaptive refinement of first-order models. In Section 4, we empirically show how this framework substantially outperforms the state-of-the-art.

## 3 CERTIFICATION OF $\epsilon$-CLOSENESS

We now introduce our verification approach that leverages local first-order models of $f(x)$, thereby enabling the application of modern techniques for neural network verification. To perform the verification, we will seek to prove that no counterexample against $\epsilon$-closeness exists. Thus we seek an assignment of the negation of our desired specification, *i.e.*,

$$\exists x : \underbrace{x \in \mathcal{X} \wedge \|f(x) - N(x)\| > \epsilon}_{\phi}. \tag{4}$$

If we find an assignment for $x$ such that the formula $\phi$ is *satisfiable*, then we can establish that $N$ is **not** a valid neural abstraction for a given accuracy $\epsilon$. As the search for satisfying assignments is exhaustive, failure to find an assignment constitutes a proof that no such assignment exists, and thus $N$ is a valid neural abstraction for a given accuracy $\epsilon$. The formula in Equation (4) is a predicate logic

(first-order logic) formula, conventionally requiring an SMT solver for verification. The standard setting for neural network verification (Brix et al., 2024) considers reasoning over inclusion properties (propositional logic), making their application to this verification task non-trivial.

In Section 3.1, we will describe the first-order models and how they can be used in the context of verification, followed by certificate refinement in Section 3.2 to combat conservatism introduced by the first-order models.

**Remark.** *The selection of $\epsilon$ can be performed empirically, based on the maximum error observed during training, or predefined according to strict application requirements. Notably, our proposed approach allows for an efficient search for the optimal $\epsilon$ within a given computational budget.*

## 3.1 FIRST-ORDER MODELS

Given that the function $f$ may include nonlinear terms, finding a satisfying assignment of $\phi$ in Equation (4) typically requires reasoning over quantifier-free nonlinear real arithmetic formulae. This is computationally challenging and does not scale efficiently with problem complexity or the number of optimization variables. To address this, we introduce an over- and under-approximation of the dynamics of $f$ using local first-order Taylor expansions of the vector field $f$. The choice of first-order models is a balance in the trade-off between expressivity, since we are verifying input-output relations, and maintaining linearity to enable formal verification.

We will adaptively partition the domain of interest $\mathcal{X}$ into hyperrectangles, which are represented as weighted $L_\infty$-balls.

**Definition 2.** *Given a hyperrectangle with radius $\delta \in \mathbb{R}^n_{\geq 0}$ and center $c \in \mathbb{R}^n$, the weighted $L_\infty$-ball around $c$, denoted by $\mathcal{H}_\delta(c)$, is defined as*

$$\mathcal{H}_\delta(c) = \{x \in \mathbb{R}^n : |x - c| \leq \delta\},$$

*where $|\cdot|$ is the element-wise absolute value and $\leq$ is interpreted element-wise. Similarly, we can define the hyperrectangle by its lower and upper corners, $\mathcal{H}^{\min} = c - \delta$ and $\mathcal{H}^{\max} = c + \delta$, respectively, as $\mathcal{H}_\delta(c) = \{x \in \mathbb{R}^n : \mathcal{H}^{\min} \leq x \leq \mathcal{H}^{\max}\}$.*

The choice of partitioning will be discussed in more detail in the following section. For now, let us introduce the local first-order Taylor expansion, including an error bound.

**Proposition 1** (Certified first-order Taylor Expansion). *Let $f : \mathbb{R}^n \to \mathbb{R}^m$ be a continuously differentiable function, and let $\mathcal{H}_\delta(c)$ be a hyperrectangle centered at $c \in \mathbb{R}^n$ with radius $\delta$. Then, there exists a hyperrectangle $\mathcal{R} \subseteq \mathbb{R}^m$ such that for all $x \in \mathcal{H}_\delta(c)$, the following relation holds:*

$$f(x) \in (f(c) + \nabla f(c)(x - c)) \oplus \mathcal{R},$$

*where $\oplus$ denotes the Minkowski sum.*

Computing $\mathcal{R}$ can be done efficiently when $f$ is twice continuously differentiable using the Lagrange error bound (see Appendix D.1 for details). The proof follows directly from Taylor's theorem for multivariate functions, along with the Lagrange error bound for higher-order terms (Joldes, 2011). For the remainder of the paper, we use the subscript indices $i \in \{1, \ldots, n\}$ when referring to the input dimensions of the function or the neural network, and $j \in \{1, \ldots, m\}$ when referring to the output dimensions. The first-order Taylor expansion in Proposition 1 provides the following sufficient condition for a valid neural abstraction.

**Theorem 1.** *Let $f : \mathbb{R}^n \to \mathbb{R}^m$, $\mathrm{N} : \mathbb{R}^n \to \mathbb{R}^m$, and let $\mathcal{H}_\delta(c) \subset \mathbb{R}^n$ be a hyperrectangle centered at $c$ with radius $\delta$. Let $f(c) + \nabla f(c)(x - c) \oplus \mathcal{R}$ be a certified Taylor expansion for $f$ in $\mathcal{H}_\delta(c)$. If for each output dimension $j \in \{1, \ldots, m\}$, there does not exist a state $x$ such that either of the following inequalities is satisfied:*

$$x \in \mathcal{H}_\delta(c) \wedge f_j(c) + \nabla f_j(c) \cdot (x - c) + \mathcal{R}_j^{\max} - \mathrm{N}_j(x) \geq \epsilon, \tag{5a}$$

$$x \in \mathcal{H}_\delta(c) \wedge \mathrm{N}_j(x) - f_j(c) - \nabla f_j(c) \cdot (x - c) - \mathcal{R}_j^{\min} \geq \epsilon, \tag{5b}$$

*then $\mathrm{N}$ is an $\epsilon$-accurate neural abstraction of $f$ over $\mathcal{H}_\delta(c)$, i.e., $\|f(x) - \mathrm{N}(x)\| \leq \epsilon, \forall x \in \mathcal{H}_\delta(c)$.*

The proof of Theorem 1 is provided in Appendix C. Both $f$ and $\nabla f$ are evaluated at the center point, $c$, of the hyperrectangle $\mathcal{H}_\delta(c)$, ensuring that all terms highlighted in orange in Equations (5a)

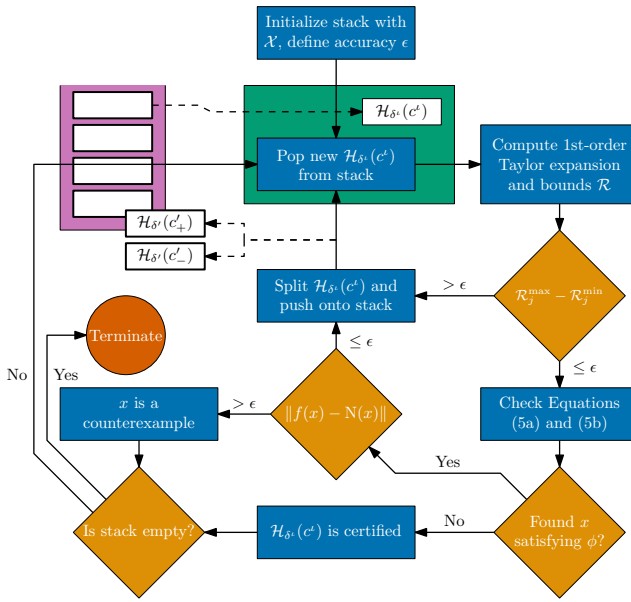

Figure 1: Graphical representation of the neural abstraction verification procedure with certificate refinement.

and (5b) remain fixed for a given hyperrectangle. In contrast, only the terms in blue vary with the specific choice of $x \in \mathcal{H}_\delta(c)$. As a result, the expression $f_j(c) + \nabla f_j(c)(x - c) + \mathcal{R}_j^{\max/\min}$ becomes linear. This allows Theorem 1 to be applied as a relaxation of the nonlinear predicate $\phi$ from Equation (4), thereby enabling formal verification to proceed without relying on SMT solvers capable of reasoning over nonlinear real arithmetic. Specifically, we employ Marabou 2.0 (Wu et al., 2024), which implements an extension of the Simplex algorithm that was originally developed to solve linear programs (Dantzig, 2002), to verify the satisfiability of Equations (5a) and (5b).

Since the domain $\mathcal{X}$ can be over-approximated by a finite union of hyperrectangles, *i.e.*, $\mathcal{X} \subseteq \bigcup_{\iota=1}^{I} \mathcal{H}_{\delta^\iota}(c^\iota)$, we can perform verification locally within each hyperrectangle $\mathcal{H}_{\delta^\iota}(c^\iota)$. By applying a local first-order Taylor expansion within each region and bounding the remainder using $\mathcal{R}^{\max}$ and $\mathcal{R}^{\min}$, we obtain tighter bounds on the approximation error compared to approximating over the full domain $\mathcal{X}$. This localized approach effectively reduces the conservatism introduced by the approximation, *i.e.*, omitting higher-order derivative terms, while maintaining soundness of the verification process.

**Remark.** *We could allow the bound $\epsilon$ to vary over the domain $\mathcal{X}$, selecting different values of $\epsilon$ for each partition $\mathcal{H}_{\delta^\iota}(c^\iota)$. This would lead to a state-dependent disturbance in Equations (1) and (2). Similarly, we could allow different $\epsilon$ for each output dimension,* i.e., *output-weighted $\epsilon$-closeness. However, for the sake of clarity and simplicity in the exposition, we omit this variation of $\epsilon$.*

### 3.2 CERTIFICATE REFINEMENT

In the previous section, we introduced first-order Taylor expansions to derive conservative over- and under-approximations of the dynamics $f$. These approximations, captured in Equations (5a) and (5b), consider the worst-case realizations of the error term $r \in \mathcal{R}$. Consequently, when we compute the bounds $\mathcal{R}_j^{\max}$ and $\mathcal{R}_j^{\min}$, the counterexample $x$ found may not always satisfy the formula $\phi$ from Equation (4). This happens because the error bounds derived from the Taylor expansion may be overly conservative. To address this issue, we propose a refinement strategy that partitions each hyperrectangle locally, enabling tighter approximations of the dynamics and reducing conservatism. The certification and partitioning strategy is illustrated in Figure 1. The decision to partition a hyperrectangle into two separate hyperrectangles, referred to as a split, results from one of two conditions:

1. the Taylor remainder term is too conservative, *i.e.*, $\mathcal{R}_j^{\max} - \mathcal{R}_j^{\min} > \epsilon$,

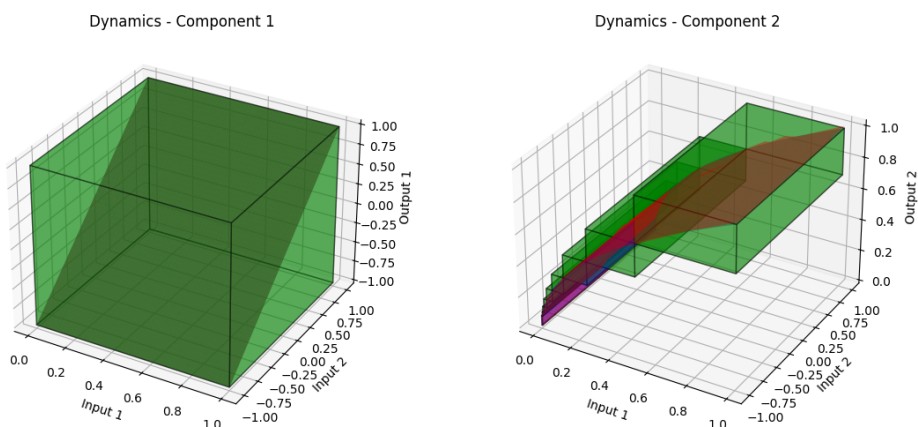

Figure 2: Partitioning of the domain to reduce conservatism. (*Left*) Linear terms do not require partitioning, as they are captured accurately by the first-order model. (*Right*) In regions of high nonlinearity (steeper dark function), finer rectangular partitioning reduces first-order approximation error by adaptively refining the domain.

  2. a counterexample $x$ does not satisfy $\|f(x) - \mathrm{N}(x)\|_j > \epsilon$.

In the first case, we split the hyperrectangle based purely on the conservatism of the Taylor approximation, which can be done without reasoning over the neural network, while in the second case, the decision to split depends on the outcome of the network verification. If $x$ satisfies $\|f(x) - \mathrm{N}(x)\| > \epsilon$, no further splitting is necessary and $x$ is returned as a proper counterexample.

When splitting a hyperrectangle, it is necessary to determine along which axis the split should occur. We choose this axis by prioritizing input dimensions according to their contribution to the Taylor approximation error in the output. Specifically, we first identify input dimensions that appear in nonlinear terms of the output of interest, $f_j(x)$, by analysing the dependency graph of $f$. Then, for each of those dimensions $i$, we evaluate their contribution to the approximation error by perturbing the center point $c$ of the hyperrectangle along that dimension. The perturbed point, denoted $c'$, is defined such that $c'_l = c_l$ for all $l \neq i$, and $c'_i = c_i + h_i$, where $h_i \in (0, \delta_i)$ is a small, fixed perturbation magnitude. For each such perturbation, we evaluate the absolute error between the first-order Taylor model and the true dynamics $f$ at $c'$. While the splitting strategy prioritizes axes based on their contribution to the Taylor remainder, the procedure is relaxed to eventually split on all axes that enter $\mathrm{N}_j(x)$ nonlinearly, as this can impact the execution time of Marabou. For the selected input dimension $i$, we split the original hyperrectangle $\mathcal{H}_\delta(c)$ into two smaller hyperrectangles. These are centered at $c + (\delta - \delta')$ and $c - (\delta - \delta')$, respectively, where the new radius vector $\delta'$ is defined elementwise as:

$$\delta'_l = \begin{cases} \delta_l & \text{if } l \neq i, \\ \frac{\delta_l}{2} & \text{if } l = i. \end{cases}$$

This effectively halves the size of the hyperrectangle along the selected axis $i$, while keeping the width in all other dimensions unchanged. Since each input dimension $x_i$ for $i \in \{1, \ldots, n\}$ can influence each output dimension $f_j(x)$ for $j \in \{1, \ldots, m\}$ differently, we perform verification and refinement separately for each output dimension.

The proposed partitioning strategy adapts the size of the hyperrectangles locally according to the nonlinearity of the function $f(x)$ based on the dependency graph and the perturbed first-order Taylor remainder. As illustrated in Figure 2, for the component $f_1(x)$, which is linear, a single hyperrectangle suffices to certify the $\epsilon$-closeness over the entire domain $\mathcal{X}$. In contrast, the second component, $f_2(x)$, contains highly nonlinear terms that necessitate finer partitioning in regions where linear approximations are no longer sufficiently tight. For many real-world systems, this targeted approach avoids the worst-case exponential growth and scales far more effectively than methods that cannot leverage this decoupling. To illustrate this, consider the Jet Engine dynamics in Appendix A.2. The dynamics of $\dot{x}$ are coupled, yet the nonlinearity only appears in $x$, while the dynamics in $\dot{y}$ are linear. Our certification refinement strategy capitalizes on this, resulting in fast and efficient verification.

Since the verification of $\epsilon$-closeness can be performed locally over partitions $\mathcal{H}_\delta(c)$, we exploit this structure to parallelize the procedure across multiple processors, significantly improving performance. To facilitate parallel execution, we employ a shared stack accessed by a pool of worker processes. The domain $\mathcal{X}$ is initially partitioned into a set of hyperrectangles, which are pushed onto the stack. Each process draws a hyperrectangle from the stack, performs the verification procedure described earlier, and either (i) certifies the region, (ii) marks it as uncertifiable if a counterexample is found, or (iii) splits the region as previously discussed. In the case of a split, the resulting subregions are pushed onto the stack. The process then retrieves the next hyperrectangle and repeats the procedure, as summarized in Figure 1.

**Remark.** *The use of a stack (Last-In-First-Out) instead of a queue (First-In-First-Out) corresponds to a depth-first rather than breadth-first exploration of the verification space, consistent with strategies from branch-and-bound algorithms (Morrison et al., 2016). A queue would be equally valid, though it would require more memory. If early termination with counterexamples is desired rather than verification until full coverage,* e.g.*, in the context of Counter-Example Guided Inductive Synthesis (Abate et al., 2018; 2022), a priority queue can be employed where the hyperrectangles would be weighted by the error relative to their volume (Lebesgue measure).*

## 4 EXPERIMENTAL RESULTS

We empirically evaluate our approach on the benchmarks introduced in Abate et al. (2022) and described in Table 1, and whose detailed dynamics can be found in Appendix A, as well as on new benchmarks designed to demonstrate the extended capabilities of our method. In particular, in what follows, we first present an empirical comparison with the method introduced in Abate et al. (2022) and based on dReal (Gao et al., 2013), which currently represents the state-of-the-art for neural abstractions; then, to highlight the generality and scalability of our approach, we include two particularly challenging tasks: (i) a neural network compression benchmark, where a network with 5 layers and 1024 neurons per layer is compressed to a network with 5 layers and 128 neurons per layer, achieving a 98.4% reduction in size; and (ii) a verification benchmark based on a trajectory prediction network introduced in Dey and Davis (2023); Lusch et al. (2018), which learns to approximate nonlinear system dynamics through Koopman operator theory. These two benchmarks are discussed in Section 4.3 and 4.2, respectively. All experiments were executed on an Intel i7-6700k CPU (8 cores) with 16GB memory.

### 4.1 COMPARISON WITH DREAL-BASED APPROACHES

To benchmark our method against the state-of-the-art, we compare it with the approach of Abate et al. (2022), which is based on dReal (Gao et al., 2013); an SMT-solver over nonlinear real arithmetic[2]. As evident from the results in Table 1, our method scales to larger models (7D) more effectively than verification using dReal. This improved scalability arises as dReal is reasoning over nonlinear real arithmetic, while our method avoids this by reasoning over local linear approximations. Our approach successfully certifies all models with the 3x[64] architecture, while dReal exceeds the 1-hour timeout on all large models, with the exception of the *WaterTank* and *NonlinearOscillator*. Our method nevertheless achieves a noticeable speedup on all models (e.g. $\approx$820x faster for the *WaterTank* experiment).

### 4.2 TRAJECTORY-LEVEL REASONING THROUGH KOOPMAN OPERATORS

We now consider abstractions for discrete-time nonlinear systems. Instead of limiting the abstraction to predicting a single next state, however, we extend its task to predicting an entire trajectory—a sequence of future states from an initial condition. To facilitate trajectory-level reasoning, we shift to an operator-theoretic viewpoint of dynamical systems, wherein the evolution of a system is described through the action of a (linear) operator on measurement functions. This framework, known as *Koopman theory*, offers a powerful lens for analysing complex, nonlinear systems (Brunton et al.,

---

[2]Note that our approach allows one to establish a certified subset of the domain, while the method in Abate et al. (2022) provides only a single counterexample. One application of partial certification is to cordon off uncertified regions and restrict the system to known, correct behaviours. Our approach can be further extended with a variable $\epsilon$ over the domain of interest, allowing for a tighter certification in general.

Table 1: Verification Results for Learning Dynamical Systems[2]

| Model | Network | Dim | Our approach | | dReal | |
|---|---|---|---|---|---|---|
| | | | Certified (%) | Time (s) | Result | Time (s) |
| WaterTank | [12] | 1 | 100.0 | 0.02 | ✓ | 0.02 |
| | 3x[64] | 1 | 100.0 | 0.56 | ✓ | 458.91 |
| JetEngine | [10, 16] | 2 | 100.0 | 4.35 | ✓ | 27.18 |
| | 3x[64] | 2 | 100.0 | 19.27 | Timeout (1h) | |
| SteamGovernor | [12] | 3 | 100.0 | 0.18 | ✓ | 39.37 |
| | 3x[64] | 3 | 100.0 | 69.47 | Timeout (1h) | |
| Exponential | 2x[14] | 2 | 100.0 | 0.23 | ✓ | 9.99 |
| | 3x[64] | 2 | 100.0 | 3.92 | Timeout (1h) | |
| NonLipschitzVectorField1 | [10] | 1 | 100.0 | 0.03 | ✓ | 0.03 |
| | 3x[64] | 1 | 100.0 | 2.14 | Timeout (1h) | |
| NonLipschitzVectorField2 | [12, 10] | 2 | 100.0 | 0.08 | ✓ | 4.55 |
| | 3x[64] | 2 | 100.0 | 11.93 | Timeout (1h) | |
| VanDerPolOscillator | 3x[64] | 2 | 100.0 | 48.76 | Timeout (1h) | |
| Sine2D | 3x[64] | 2 | 100.0 | 69.06 | Timeout (1h) | |
| NonlinearOscillator | 3x[64] | 1 | 100.0 | 0.35 | ✓ | 234.52 |
| LowThrustSpacecraft | 3x[64] | 7 | 100.0 | 94.51 | Timeout (1h) | |

2022). Notably, Koopman theory provides a route to uncovering intrinsic coordinate systems in which the nonlinear dynamics manifest as linear. Originally introduced in Koopman (1931), the Koopman operator represents a nonlinear dynamical system via an infinite-dimensional linear operator acting on a Hilbert space of measurement functions. Despite the underlying system's nonlinearity, the Koopman operator is linear, and its spectral decomposition fully characterizes the system's behaviour (Brunton et al., 2016; 2022; Korda and Mezić, 2018).

In general, the Koopman operator is infinite-dimensional, making its exact computation intractable. Thus, the aim is commonly to construct finite-dimensional approximations that capture the dominant behaviour of the system. This entails identifying a low-dimensional invariant subspace spanned by eigenfunctions of the Koopman operator, within which the dynamics evolve linearly. Despite the promise of Koopman embeddings, obtaining tractable representations has remained a central challenge in control theory. Utilizing neural networks to discover and represent Koopman eigenfunctions has emerged as a promising approach in recent years (Lusch et al., 2018; Dey and Davis, 2023). While Koopman operators are commonly learned from data and a typical analysis of learned Koopman embeddings would verify structural properties—such as the orthonormality of eigenfunctions in Hamiltonian systems— such a treatment lies beyond the scope of this work. We instead demonstrate that our approach to verification can be applied to neural architectures deployed for learning Koopman embeddings, by verifying that the learned system evolution remains $\epsilon$-close to the true system dynamics.

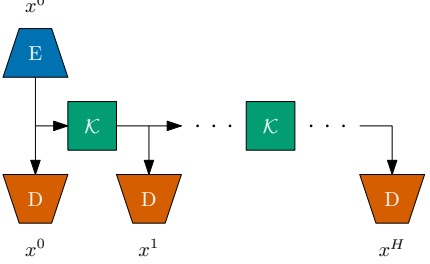

Figure 3: The network architecture used to learn a Koopman operator. The Encoder (blue) lifts the input into a higher dimensional space where linear multiplication with $\mathcal{K}$ (green) propagates the system. To obtain trajectory points, $[x^0, x^1, \dots, x^{50}]$, each propagated state in the lifted space is decoded (orange).

We adopt a standard setup using the dlkoopman library (Dey and Davis, 2023). The network architecture comprises an autoencoder that learns the encoding and decoding of states into a Koopman-invariant subspace, along with a linear transformation within that subspace (Figure 3). This network architecture can be interpreted as a discrete-time neural abstraction that advances the system ahead one-time step and outputs the state at time step $k + 1$ (Equation (2)). Thus, the output of the network

is a trajectory, *i.e.*, a sequence of $H$-subsequent states. Applications of such trajectory tasks are commonly found throughout control theory, notably in model predictive control (Rawlings et al., 2017).

We consider the Quadratic System provided in Appendix A.12, a benchmark problem frequently studied in the literature (H. Tu et al., 2014; Brunton et al., 2016; Lusch et al., 2018; Dey and Davis, 2023). To improve reproducibility, we leverage the dataset provided by Dey and Davis (2023) to learn the evolution of the system from data. The final trained model takes an initial state $x^0$ and produces the sequence $[x^0, x^1, \ldots, x^{50}]$, which represents the evolution of the dynamics. The network architecture is summarized in Figure 3. Verification of the abstraction completed in 162.60 seconds with 29 counterexamples found and $99.03\%$ of the domain certified. An interesting observation is that, although the network achieved a low prediction loss—specifically, a mean squared error of 0.002 on the validation set—our verification framework was still able to identify counterexamples where the prediction error exceeds the specified tolerance of $\epsilon = 0.1$. At the same time, the method certifies that the network $\epsilon$-accurately predicts the system evolution over $99.03\%$ of the input domain. Recall that in the presence of counterexamples, previous verification methods fail to identify regions where the model remains $\epsilon$-accurate. Meanwhile, our approach offers valuable insight by localizing the regions in which the model can still be trusted, even when global verification fails. We provide analysis and further discussion on the counterexamples in Appendix E.

### 4.3 COMPRESSION OF LEARNED DYNAMICS

To demonstrate the capabilities of our approach beyond constructing abstractions of known analytical dynamics, we apply the verification procedure to a neural network compression benchmark. This not only serves to demonstrate our approach's ability to handle networks with a large number of parameters but also showcases its broader applicability beyond the dynamical systems and control literature.

State-of-the-art techniques often produce large, over-parameterized neural networks. While highly accurate and implicitly regularized (Martin and Mahoney, 2021; Belkin et al., 2019; Jacot et al., 2018), these models present two major drawbacks that motivate the need for knowledge distillation: they are computationally expensive, which is problematic for applications like embedded systems, and they are difficult to analyze, *e.g.*, via the Piece-Wise Affine (PWA) representation induced by their ReLU structure Gou et al. (2021). Neural network compression aims to mitigate this by reducing model size while preserving input-output behaviour (Luo et al., 2017; Memmel et al., 2024). The key challenge, which motivates this benchmark, is formally guaranteeing that the compressed network stays $\epsilon$-close to the original.

For this compression benchmark, we first train a 5-layer ReLU network with 1024 neurons per layer to learn the dynamics of the Lorenz attractor (described in Appendix A.10) from observed trajectories. Using a simple teacher–student architecture Gou et al. (2021), we reduce the model by independently training a smaller ReLU network, consisting of 5 layers with 128 neurons per layer, to replicate the input-output behaviour of the larger model—without access to its training trajectories or the underlying system dynamics. The larger model comprises $4,205,571$ parameters, while the compressed model contains only $66,947$, corresponding to a $98.4\%$ reduction in size. From the perspective of verification, the larger network serves as the reference dynamics, while the smaller network acts as an abstraction of those learned dynamics. This setup allows us to evaluate our method's ability to handle large-scale networks and non-analytical dynamics.

To fit within our verification framework, we construct certified linearizations of the neural network dynamic model using CROWN (Zhang et al., 2022). This is necessary since the network is not twice continuously differentiable, which is required to calculate Lagrange error bounds. CROWN computes (local) linear relaxations of a nonlinear function, particularly neural networks, by recursively relaxing nonlinearities on the computation graph corresponding to the function (see Appendix D.2 for details). The verification procedure was executed with $\epsilon = 0.6$ and completed in 89 hours and 13 minutes. In total, $3,504,327$ hyperrectangles were checked and certified or further split according to the algorithm in Figure 1.

## 5 CONCLUSION

We presented a method for certifying neural abstractions of dynamical systems using a parallelisable domain partitioning strategy in conjunction with local first-order models. This allows us to efficiently verify complex nonlinear systems without relying on expensive SMT solvers required to reason over nonlinear arithmetic. To address the fundamental limitation on scalability, our certificate refinement strategy verifies each output dimension independently. This approach ensures that local refinement is only performed on input dimensions that significantly affect a given output in a nonlinear manner. We demonstrated the effectiveness of our approach on several new challenging benchmarks, including network compression tasks and the verification of a trajectory prediction task based on Koopman linearization.

## 6 REPRODUCIBILITY STATEMENT

All experiments can be reproduced using the scripts provided in the accompanying codebase available at `https://anonymous.4open.science/r/certified-neural-approximations-E679`. The repository contains code for the dynamics, training, and verification, pre-trained models stored as .onnx file, and a Docker image for a reproducible environment. The randomness in all experiments controlled by explicitly setting seed for the pseudorandom number generators. The hyperparameters are listed in Appendix B.

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

# A  DYNAMICAL SYSTEMS

## A.1  WATER TANK

A simple first-order nonlinear dynamical system modelling the water level in a tank with constant inflow and outflow dependent on the water pressure (proportional to the square root of height).

$$\dot{x} = 1.5 - \sqrt{x}$$

where $x > 0$ represents the water level. For certification, we use $\epsilon = 0.097$ for the small network and a tighter $\epsilon = 0.007$ for the larger network. The input domain for certification is $\mathcal{X} = [0.1, 10.0]$.

## A.2  JET ENGINE

A two-dimensional nonlinear system with polynomial dynamics that models the behaviour of a simplified jet engine:

$$\dot{x} = -y - 1.5x^2 - 0.5x^3 - 0.1$$
$$\dot{y} = 3x - y$$

For certification, we use $\epsilon = 0.039$ for the small network and a tighter $\epsilon = 0.012$ for the larger network. The input domain for certification is $\mathcal{X} = [-1.0, 1.0] \times [-1.0, 1.0]$.

## A.3  STEAM GOVERNOR

A mechanical governor used in steam engines, formulated as a three-dimensional nonlinear system with trigonometric terms:

$$\dot{x} = y$$
$$\dot{y} = z^2 \sin(x) \cos(x) - \sin(x) - 3y$$
$$\dot{z} = -(\cos(x) - 1)$$

For the implementation we use the trigonometric identity $\sin(x)\cos(x) = \frac{1}{2}\sin(2x)$ to rewrite $\dot{y}$ as $\frac{1}{2}z^2\sin(2x) - \sin(x) - 3y$. For certification, we use $\epsilon = 0.105$ for the small network and a tighter $\epsilon = 0.06$ for the larger network. The input domain for certification is $\mathcal{X} = [-1.0, 1.0] \times [-1.0, 1.0] \times [-1.0, 1.0]$.

## A.4  EXPONENTIAL SYSTEM

The exponential system features highly nonlinear dynamics with nested nonlinearities combining trigonometric, exponential, and polynomial terms:

$$\dot{x} = -\sin(e^{y^3+1}) - y^2$$
$$\dot{y} = -x$$

For certification, we use $\epsilon = 0.112$ for the small network and a tighter $\epsilon = 0.04$ for the larger network. The input domain for certification is $\mathcal{X} = [-1.0, 1.0] \times [-1.0, 1.0]$.

## A.5  NON-LIPSCHITZ VECTOR FIELD 1 (NL1)

A non-Lipschitz continuous vector field:

$$\dot{x} = y$$
$$\dot{y} = \sqrt{x}$$

where $x \geq 0$. For certification, we use $\epsilon = 0.11$ for the small network and a tighter $\epsilon = 0.03$ for the larger network. The input domain for certification is $\mathcal{X} = [0.0, 1.0] \times [-1.0, 1.0]$.

## A.6 Non-Lipschitz Vector Field 2 (NL2)

A more challenging non-Lipschitz continuous vector field:

$$\dot{x} = x^2 + y$$
$$\dot{y} = (x^2)^{1/3} - x$$

For certification, we use $\epsilon = 0.081$ for the small network and a tighter $\epsilon = 0.02$ for the larger network. The input domain for certification is $\mathcal{X} = [-1.0, 1.0] \times [-1.0, 1.0]$.

## A.7 Van der Pol Oscillator

The classical Van der Pol oscillator with nonlinear damping:

$$\dot{x}_1 = x_2$$
$$\dot{x}_2 = \mu(1 - x_1^2)x_2 - x_1$$

where $\mu > 0$. For certification, we use $\epsilon = 0.25$. The input domain for certification is $\mathcal{X} = [-3.0, 3.0] \times [-3.0, 3.0]$.

## A.8 Sine 2D System

The Sine 2D system represents a two-dimensional nonlinear oscillator with sinusoidal coupling:

$$\dot{x} = \sin(\omega_y \cdot y)$$
$$\dot{y} = -\sin(\omega_x \cdot x)$$

with parameter values $\omega_x = 1.0, \omega_y = 0.5$. For certification, we use $\epsilon = 0.02$. The input domain for certification is $\mathcal{X} = [-\pi, \pi] \times [-\pi, \pi]$. We utilize a LeakyReLU activation function for networks learning the Sine 2D System, both to improve learning accuracy and to demonstrate the applicability of our approach beyond standard ReLU activation functions.

## A.9 Nonlinear Oscillator

The nonlinear oscillator combines linear, cubic, and sinusoidal terms:

$$\dot{x} = -ax - bx^3 + c\sin(x)$$

with parameter values $a = 1.0, b = 1/2, c = 0.3$. For certification, we use $\epsilon = 0.165$. The input domain for certification is $\mathcal{X} = [-3.0, 3.0]$.

## A.10 Lorenz Attractor

The three-dimensional Lorenz Attractor, famous for exhibiting chaotic behaviour:

$$\dot{x} = \sigma(y - x)$$
$$\dot{y} = x(\rho - z) - y$$
$$\dot{z} = xy - \beta z$$

with parameter values $\sigma = 10$, $\rho = 28$, and $\beta = 8/3$. The input domain for certification is $\mathcal{X} = [-30.0, 30.0] \times [-30.0, 30.0] \times [0.0, 60.0]$

## A.11 LOW THRUST SPACECRAFT

The dynamics of spacecraft with continuous low-thrust propulsion on a planar orbit around Earth. The system has 5 states $(r, \theta, v_r, v_\theta, \Delta m)$ and 2 control inputs $(T, \alpha)$.

$$\dot{r} = v_r$$
$$\dot{\theta} = \frac{v_\theta}{r}$$
$$\dot{v}_r = -\frac{\mu}{r^2} + \frac{v\theta^2}{r} + \frac{T \cdot \cos(\alpha)}{m_0 + \Delta m}$$
$$\dot{v}_\theta = -\frac{v_r \cdot v\theta}{r} + \frac{T \cdot \sin(\alpha)}{m_0 + \Delta m}$$
$$\dot{m} = -\frac{T}{v_{\text{exhaust}}}$$

where:

- $r$ is the radial distance from the central body
- $\theta$ is the azimuthal angle
- $v_r$ is the radial velocity component
- $v_\theta$ is the tangential velocity component
- $\Delta m$ is the propellant mass
- $T$ is the magnitude of the thrust
- $\alpha$ is the angle of the applied thrust
- $\mu$ is the gravitational parameter of the central body
- $m_0$ is the initial spacecraft mass
- $v_{\text{exhaust}}$ is the propellant exhaust velocity

## A.12 QUADRATIC SYSTEM DYNAMICS AND DISCRETE-TIME SOLUTION DERIVATION

A simple system governed by the continuous time dynamics:

$$\dot{x}_1 = \mu x_1$$
$$\dot{x}_2 = \lambda(x_2 - x_1^2)$$

where $x_1$ and $x_2$ are the state variables, and $\mu$ and $\lambda$ are system parameters. The system includes a linear term for $x_1$ and a quadratic term involving $x_1^2$ in the equation for $x_2$. For training, the initial conditions there chosen at random and the trajectory is computed over the time horizon $[0, 1]$, with parameters $\mu = -0.05, \lambda = -1$ and timestep of $0.02$. The input domain for certification is $\mathcal{X} = [-0.5, 0.5] \times [-0.5, 0.5]$.

To generate trajectories of the system, we integrate the system numerically and sample the trajectory evenly across the time horizon. For the purpose of verification, we derive the analytic expression of the discrete-time system. The differential equation for $x_1(t)$ yields the solution

$$x_1(t) = x_1(0)e^{\mu t}$$

Solving the differential equation for $x_2(t)$ with $2\mu \neq \lambda$ yields

$$x_2(t) = \left(x_2(0) + \frac{\lambda x_1(0)^2}{2\mu - \lambda}\right)e^{\lambda t} - \frac{\lambda x_1(0)^2}{2\mu - \lambda}e^{2\mu t}$$

For a discrete time step $\Delta t$, we obtain the discrete-time system is:

$$x_1^{n+1} = x_1^n e^{\mu \Delta t}$$
$$x_2^{n+1} = \left(x_2^n + \frac{\lambda(x_1^n)^2}{2\mu - \lambda}\right)e^{\lambda \Delta t} - \frac{\lambda(x_1^n)^2}{2\mu - \lambda}e^{2\mu \Delta t}$$

where $\Delta = 0.02$ in the experiments.

## B    HYPERPARAMETERS

For all experiments, the architecture is described in Section 4 and the networks are trained with the loss function defined in Section 2.1 using the AdamW optimizer with a weight decay of $1e{-}4$. The gradient is clipped to a norm of 1 if the norm exceeds this limit.

For the experiment comparing the proposed framework with dReal, the learning rate is initialized at $1e{-}3$ and reduced according to a cosine annealing schedule to a minimum of $1e{-}6$ over $50\,000$ iterations. The data is sampled at each iteration uniformly over the domain $\mathcal{X}$, with a batch size of 4096.

For the compression benchmark, the two networks are trained with different parameters. First, the large-scale network is trained from steps of the Lorenz attractor with a discrete time step $\Delta t = 0.02$, obtained as trajectories using an RK45 integrator for 32 time steps and 128 different initial conditions in $\mathcal{X}$, for a batch size of 4096. The learning rate is initialised to $1e{-}6$ and reduced by a factor of 0.9 when encountering a loss plateau for 2000 iterations. The network is trained for $500\,000$ iterations. The compressed network is trained with a fixed learning rate of $1e{-}6$ for $1\,000\,000$ iterations where data is sampled at each iteration uniformly over the domain $\mathcal{X}$, with a batch size of 4096.

For the Koopman benchmark, the network is trained for 200 epochs over a dataset of 10500 trajectories, with a batch size of 125, and with a weight decay of $1e{-}6$.

## C    PROOF OF THEOREM 1

*Proof.* We can express $f(x)$ as:
$$f(x) = f(c) + \nabla f(c)(x - c) + r,$$
where $r \in [\mathcal{R}^{\min}, \mathcal{R}^{\max}]$. If no satisfying assignment exists for Equation (5a), it follows that:
$$f_j(c) + \nabla f_j(c)(x - c) + \mathcal{R}_j^{\max} - \mathrm{N}_j(x) < \epsilon.$$
Since $f_j(c) + \nabla f_j(c)(x - c) + \mathcal{R}_j^{\max}$ provides an upper bound for $f_j(x)$, we have:
$$f_j(x) - \mathrm{N}_j(x) < \epsilon.$$
Similarly, for the lower bound, Equation (5b):
$$\mathrm{N}_j(x) - f_j(x) < \epsilon.$$
These bounds hold for all $j \in \{1, \ldots, m\}$. Therefore, when no satisfying assignment is found for all $j \in \{1, \ldots, m\}$, it follows that:
$$\|f(x) - \mathrm{N}(x)\| \leq \epsilon, \quad \forall x \in \mathcal{H}_\delta(c).$$
$\square$

## D    CERTIFIED LINEARIZATIONS

### D.1    CERTIFIED TAYLOR EXPANSIONS OF ELEMENTARY FUNCTIONS

Suppose that $f : \mathbb{R}^n \to \mathbb{R}^m$ is composed of smooth elementary functions and is at least twice continuously differentiable. We consider the first-order Taylor approximation of $f$ around a point $x_0 \in \mathbb{R}^n$:
$$f(x) \approx f(x_0) + J_f(x_0)(x - x_0),$$
where $J_f(x_0)$ is the $m \times n$ Jacobian matrix of $f$ at $x_0$. We define the remainder $\mathcal{R}(x) \in \mathbb{R}^m$ componentwise for each $j \in \{1, \ldots, m\}$:
$$\mathcal{R}_j(x) = f_j(x) - \left[ f_j(x_0) + \nabla f_j(x_0)^\top (x - x_0) \right].$$
This remainder captures the contribution of second and higher-order terms. By the *Lagrange form* of the Taylor remainder, we have:
$$\mathcal{R}_j(x) = \frac{1}{2}(x - x_0)^\top \nabla^2 f_j(\xi)(x - x_0),$$

for some $\xi$ on the line segment between $x_0$ and $x$. Here $\nabla^2 f_j(\xi)$ is the $n \times n$ Hessian matrix of the $j$-th component function. To bound the magnitude of the remainder, we find a constant $M_j$ (bounding the spectral norm of the Hessian):

$$\|\nabla^2 f_j(x)\|_2 \leq M_j \quad \text{for all } x \in \mathcal{H}_\delta(c),$$

where $\mathcal{H}_\delta(c)$ is the compact, convex hyperrectangle containing $x_0$ and $x$. Then,

$$|\mathcal{R}_j(x)| \leq \frac{1}{2} M_j \|x - x_0\|_2^2.$$

By simple application of the chain rule, we can similarly bound compositions, i.e., $f(g(x))$. Let $y = g(x)$ be the input function (where $g : \mathbb{R}^n \to \mathbb{R}^m$) with its own expansion centered at $x_0$, and $f : \mathbb{R}^m \to \mathbb{R}^m$ be the elementary function we are applying (element-wise). We define $y_0 = g(x_0) \in \mathbb{R}^m$ as the center for the expansion of $f$.

The total remainder for the composition $f(g(x))$ is a vector $\mathcal{R}_{\text{total}}(x) \in \mathbb{R}^m$:

$$\mathcal{R}_{\text{total}}(x) = \mathcal{R}_{\text{prop}}(x) + \mathcal{R}_{\text{local}}(x)$$

This splits the remainder into two parts:

1. **Propagated Remainder:** $\mathcal{R}_{\text{prop}}(x) = J_f(y_0)\mathcal{R}_g(x)$. This term propagates the remainder of the input function, $\mathcal{R}_g(x) \in \mathbb{R}^m$, via the Jacobian of $f$. Since $f$ is an element-wise function (e.g., $e^x$), its Jacobian $J_f(y_0)$ is a diagonal matrix:

$$J_f(y_0) = \text{diag}\left(f'(y_{0,1}), \ldots, f'(y_{0,m})\right)$$

2. **Local Remainder:** $\mathcal{R}_{\text{local}}(x) = \mathcal{R}_f(g(x))$. This term is the local remainder of the elementary function $f$ itself, evaluated at the input $y = g(x)$.

To produce tight bounds for $\mathcal{R}_f(y)$ (element-wise), we leverage properties over the input hyperrectangle $I_y = [y_{\min}, y_{\max}]$:

- If $f$ is **convex** on $I_y$ (i.e., $f''(y_j) \geq 0$ for all $y_j \in [y_{j,\min}, y_{j,\max}]$), the linear approximation is an underestimate. The local remainder $\mathcal{R}_f(y)$ is non-negative.

$$\mathcal{R}_{\min} = \mathbf{0}$$
$$\mathcal{R}_{\max} = \max\left(f(y_{\min}) - f_L(y_{\min}), f(y_{\max}) - f_L(y_{\max})\right)$$

- If $f$ is **concave** on $I_y$ (i.e., $f''(y_j) \leq 0$), the linear approximation is an overestimate. The local remainder $\mathcal{R}_f(y)$ is non-positive.

$$\mathcal{R}_{\min} = \min\left(f(y_{\min}) - f_L(y_{\min}), f(y_{\max}) - f_L(y_{\max})\right)$$
$$\mathcal{R}_{\max} = \mathbf{0}$$

All operations ($\max$, $\min$, $f_L$, $f$) are applied element-wise.

For functions with a known, fixed global range, such as $\sin(y)$ and $\cos(y)$ where $f(y) \in [-1, 1]$, or for monotonically increasing/decreasing functions where we can utilise the fact that the maximum/minimum of the $f(y)$ is easily found by checking the boundaries of the domain of interest, we can perform an additional, post-processing step to tighten the remainder bounds, i.e. we clip $R_f(y)$ to the domain

$$R_f(y) \in [L - f_L(y), U - f_L(y)] \subseteq \left[L - \max_y f_L(y), U - \min_y f_L(y)\right],$$

where $f(y) \in [L, U]$.

### D.2 Using CROWN

If the function $f$ is not twice continuously differentiable, e.g. for a ReLU network, then the Lagrange bound is not valid to compute the Taylor remainder. Instead, we employ CROWN (Zhang et al., 2022), also known as Linear Bound Propagation, which was originally developed to verify neural

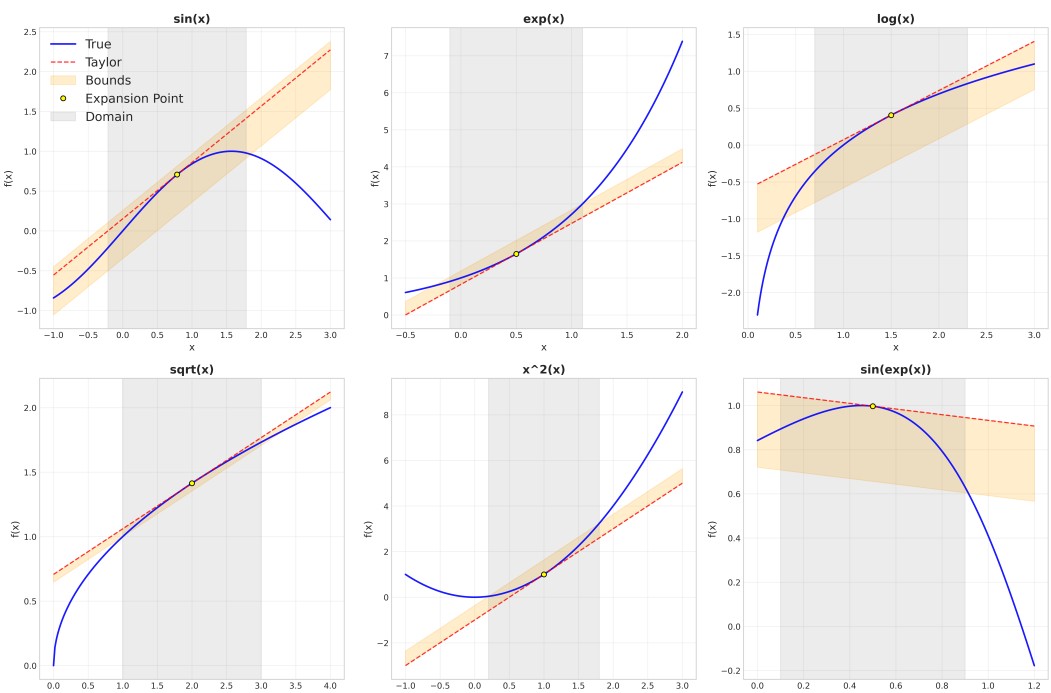

Figure 4: Taylor expansions of common elementary functions

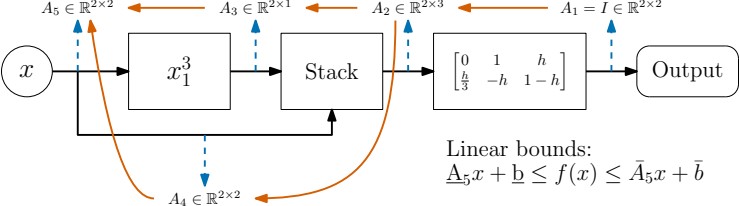

Figure 5: The computation graph with CROWN annotation for function $f(x) = \begin{bmatrix} x_1 + hx_2 \\ x_2 + h(\frac{1}{3}x_1^3 - x_1 - x_2) \end{bmatrix}$.

networks via linear relaxations of the network. CROWN comes in several flavours including forward-mode (similar to the Taylor bound propagation above), backward-mode, forward-backward-mode (relaxations via forward mode), CROWN-IBP (relaxations via Interval Bound Propagation), $\alpha$-CROWN (optimization of bounds), $\beta$-CROWN (neuron splitting branch-and-bound), and GCP-CROWN (general cutting planes). We only employ backward mode, which is the original version. In the remainder, when we refer to CROWN we mean backward-mode CROWN.

The idea of CROWN is to operate on the computation graph and relax nonlinearities based on node local input intervals, which can be computed using CROWN itself recursively. Figure 5 exemplifies this process on a composition of polynomial functions, for ease of exposition. First, the nonlinear term $x^3$ is locally relaxed to upper and lower linear functions based on the input range. Then, starting from the output with the linear functions $\underline{A}y + \underline{b} = Iy + 0$ and $\overline{A}y + \overline{b} = Iy + 0$, the bounding functions are propagated backward through the computation graph to the input. If the computation graph contains multiple nonlinearities, using the backward propagation from each nonlinearity to the input, the input range to each node is calculated to locally relax it. We refer to (Xu et al., 2020) for details on how to compute local relaxations based on input bounds and how to propagate through linear and locally relaxed nonlinear operations.

### D.3 Using Lipschitz constants

While conservative, it is possible to construct linear relaxations from the (local) Lipschitz constant of the function $f$, if such exists, and evaluation of the function at a point $c$. The approach is shown in the following constructive proof.

**Proposition 2.** *For any function $f$ locally Lipschitz in a hyperrectangle $\mathcal{H}_\delta(c) \subset \mathbb{R}^n$ with the Lipschitz constant $L_f$, there exist linear relaxations $\underline{A}x + \underline{b}$ and $\overline{A}x + \overline{b}$ of $f$ in $\mathcal{H}_\delta(c)$.*

*Proof.* We prove the statement by construction. A (local) Lipschitz constant $L_f$ of $f$ in $\mathcal{H}_\delta(c)$ implies that

$$\|f(x_1) - f(x_2)\|_\infty \le L_f \|x_1 - x_2\|_\infty, \quad \text{for all } x_1, x_2 \in \mathcal{H}_\delta(c). \tag{6}$$

This limit to the rate of change implies the following component-wise bounds

$$f(c) - L_f \|x - c\|_\infty \cdot \mathbf{1} \le f(x) \le f(c) + L_f \|x - c\|_\infty \cdot \mathbf{1}, \quad \text{for all } x \in \mathcal{H}_\delta(c), \tag{7}$$

where $\mathbf{1} \in \mathbb{R}^m$ is a vector of all ones. Let $M = \sup_{x \in \mathcal{H}_\delta(c)} \|x - c\|_\infty = \|r\|_\infty$. Then, we obtain the linear (interval) relaxations

$$\underline{b} = f(c) - L_f M \cdot \mathbf{1} \le f(x) \le f(c) + L_f M \cdot \mathbf{1} = \overline{b}, \quad \text{for all } x \in \mathcal{H}_\delta(c), \tag{8}$$

where $\underline{A} = \overline{A} = 0$. Thus, trivial linear relaxations always exist for any locally Lipschitz continuous function. $\square$

## E    Analysis of Counterexamples in the Koopman Benchmark

In Section 4.2, we reported that the Koopman verification benchmark certified 99.03% of the domain and identified 29 counterexamples. We provide a brief analysis of these counterexamples to illustrate the advantages of having access to both certified regions and a set of counterexamples. This is in contrast with the capabilities of SMT solvers, which typically provide only a single counterexample and no information about certified regions.

It is important to emphasise that a counterexample consists of an input together with a *single* output dimension in which the $\epsilon$-closeness condition is violated. As a result, the 29 counterexamples obtained in the benchmark correspond to only two distinct inputs. These two inputs violate the $\epsilon$-closeness condition across multiple output dimensions. Recall the structure of the Koopman benchmark: the input is an initial point in the state-space, while the outputs are states along a trajectory. Thus, if one state along the trajectory is erroneous, there is an increased likelihood of repeated counterexamples across output dimensions (i.e., along the trajectory) — a failure to correctly predict a trajectory corresponds to multiple states along the trajectory deviating from the ground truth values.

Further analysing the counterexamples, we note that the counterexamples lie near a corner of the state space, regions where counterexamples are most likely to arise, as such areas are often underrepresented during training.

An increased focus on corners of the state space or expanding the state space during training would likely improve the performance of the certified neural approximation. Alternatively, a larger or state-dependent epsilon could be chosen if the observed behaviour were to be considered sufficient.

