# OpenReview forum: "Certified Neural Approximations of Nonlinear Dynamics"
_ICLR.cc/2026/Conference — Submitted to ICLR 2026_

### Official Review · Reviewer_RDFo · 2025-10-17

**Soundness:** 3
**Presentation:** 4
**Contribution:** 2
**Rating:** 4
**Confidence:** 4

**Summary:**

This paper provides a method for creating neural abstractions of nonlinear functions with certified error bounds. The method involves two key features. The first is a method for abstracting the nonlinear function being approximated using linear bounds to feed into a neural network verification tool. The second is an adaptive refinement strategy to efficiently check the property over the input space. The authors compare their method to previous work on neural abstractions and show that they are able to verify properties more quickly. They also scale their method to two new problems related to Koopman operators and neural network compression.

**Strengths:**

- I very much appreciate the writing quality, the simplicity of the notation, and the figures used throughout the paper. The paper was easy to follow.
- Neural abstractions are an interesting area of study that can help verify dynamical systems and, as the authors show, verify compressed representations.
- The two new problems introduced in the paper related to Koopman operators and neural network compression could be used as benchmarks for future techniques developed for neural abstractions.

**Weaknesses:**

- The proposed methodology seems to be a very specific instance of a more general idea. The general idea is to create a linear abstraction of a nonlinear function so that it can be used to verify the neural abstraction.  In the paper, the authors discuss creating this abstraction with a first-order approximation. However, there are multiple ways to obtain linear bounds on a nonlinear function (e.g. CROWN, which was used in the neural network compression example), and it is not justified why a first-order approximation is necessary.
- The method is only tested using the Marabou neural network verification tools, and the authors do not make clear whether other verifiers could also be used (and what effect this might have on performance).
- For the neural network compression example, the authors must use CROWN rather than a first-order approximation since the original network is not continuous. This modification to the methodology is only mentioned briefly at the end of the paper, and it is not clear how it can be incorporated into the partitioning strategy introduced in section 3.2.

**Questions:**

- Can the method be extended to other neural network verification tools? How will this affect performance?
- Could this method be subbed into the method presented by Abate et al., which checks properties using the final abstraction? If so, it would be useful to include this fact in the paper.
- How does the partitioning strategy work with CROWN?
- Line 465: what does “non-analytical dynamics” mean?
- Should proposition 1 be using the Jacobian or gradient?

Suggestions:
- Line 118: not -> nor
- “the procedure is relaxed to eventually split on all axes that enter Nj(x) nonlinearly, as this can impact the execution time of Marabou.” I am not sure what this means.

---

> ### Author Response · Authors · 2025-11-18
>
> We thank the reviewer for their constructive comments. We address the reviewer’s questions and concerns in turn.
>
> ## The choice of Marabou and other verification tools
> Our choice of Marabou is primarily due to its support for general linear constraints between input and output variables. Most existing neural network verification tools (e.g., those following the VNN-LIB format used in VNN-COMP) restrict specifications to inclusion properties---verifying that the network output lies within a prescribed set given an input set. In contrast, our verification statements in Eqs. (5a) and (5b) describe *linear* surrogate relations between the input and output, which fall outside the expressivity of such tools. It is possible to combine the linear bounds of CROWN *over the network* to fully linearize the statement in Eq. (4); however, we selected Marabou as it directly supports the structure and format of our specification.
>
> ## Incorporation with the framework of Abate et al.
>
> Our work directly addresses the main scalability bottleneck in Neural Abstractions (Abate et al., 2022): the verification of $\epsilon$-closeness between a neural network and the ground-truth function. In the case of a ReLU-activated network, our approach can be directly substituted for the verification step in Abate et al. while maintaining their subsequent approach of constructing a piecewise-linear hybrid model for use in Flow*. We will clarify this point in the revision of our manuscript.
>
> Since the primary limitation in Abate et al. concerned the scalability of the verification step, this was the central focus of our work. By deriving a linear surrogate specification for the verification problem (Theorem 1) and combining it with our adaptive partition refinement scheme (Section 3.2), we achieve the same level of formal guarantees as the SMT-based approach in Abate et al., while providing a scalable, generalizable method applicable far beyond their original scope—as demonstrated by our neural network compression and trajectory-level reasoning experiments.
>
> ## Modification of partitioning strategy with CROWN
>
> When using CROWN, the partitioning strategy remains well-defined. As with Taylor-based linearizations, we perturb the center point of each hyperrectangle along every input axis and evaluate the corresponding change in the linear bound. The axis with the largest estimated error contribution—computed as the average of the upper and lower bound deviations—is selected for splitting. This ensures that the refinement mechanism operates consistently across both Taylor- and CROWN-based linear relaxations.
>
> ## Non-analytical dynamics
>
> By “non-analytical dynamics,” we refer to functions that are not analytic, such as ReLU-activated neural networks. This distinction is relevant because Taylor expansions of analytic functions converge to the function everywhere within their domain, whereas non-analytic functions (e.g., piecewise-linear or discontinuous functions) require alternative bounding techniques like CROWN to construct certified linear relaxations.
>
> ## Jacobian vs. gradient
> We thank the reviewer for their attention to detail. In this case, it is a matter of notation: the $\nabla$ operator is understood as producing the Jacobian matrix when applied to a vector-valued function. We will clarify this in the manuscript.
>
> ## Motivation for using first-order approximations
> We use first-order approximations as they provide tighter bounds than simple interval enclosures while remaining computationally tractable. Higher-order polynomial approximations could yield tighter local bounds but would transform the verification subproblems from linear programs (LPs) into quadratic or higher-order polynomial programs, which are significantly more computationally expensive. LP solvers, by contrast, scale efficiently and are well supported by modern neural network verifiers such as Marabou.
>
> Moreover, the adaptive refinement mechanism we introduce mitigates the potential conservatism of first-order models by locally tightening bounds only where necessary. In practice, this approach achieves high precision without sacrificing scalability. Our framework remains flexible and could incorporate higher-order bounding functions in future work, though this would require alternative solvers to maintain tractability.

---

> > ### Comment · Reviewer_RDFo · 2025-11-25
> >
> > I thank the authors for their clarifications. I think the paper needs restructuring to better abstract the step of obtaining linear bounds (CROWN vs. first order approx).

---

### Official Review · Reviewer_PF2z · 2025-10-29

**Soundness:** 2
**Presentation:** 1
**Contribution:** 1
**Rating:** 2
**Confidence:** 5

**Summary:**

The paper certifies NN surrogates of nonlinear dynamics via first-order Taylor models with bounded remainders, adaptive partitioning, and a linear NN verifier (Marabou). It reports strong wall-clock gains over a dReal/SMT baseline and two extra demos (Koopman trajectories, compression).

**Strengths:**

- The technique is sound in theory.

**Weaknesses:**

The paper frames “avoiding SMT over nonlinear reals” as its contrastive novelty and compares to dReal (SMT) as “state of the art”. But modern NN-controlled-system (NNCS) verification frameworks built on Taylor/polynomial models (e.g., Verisig[1], POLAR[2]) already avoid SMT by using TM/polynomial propagation. The paper does not cite or discuss them, and the References section corroborates the omission. This leads to wrong baselining and an incomplete positioning of contributions. I believe the novelty of this paper is very unconvincing before the authors provide a direct comparison with Verisig and POLAR both theoretically and empirically.

[1] Ivanov, Radoslav, et al. "Verisig 2.0: Verification of neural network controllers using taylor model preconditioning." International Conference on Computer Aided Verification. Cham: Springer International Publishing, 2021.

[2] Huang, Chao, et al. "POLAR: A polynomial arithmetic framework for verifying neural-network controlled systems." International Symposium on Automated Technology for Verification and Analysis. Cham: Springer International Publishing, 2022.

**Questions:**

- Please clarify the novelty compared with Verisig and POLAR both theoretically and empirically.

---

> ### Author Response · Authors · 2025-11-18
>
> We thank the reviewer for the comments. However, we would like to stress that while Verisig 2.0 and POLAR indeed employ Taylor or polynomial models, our approach differs fundamentally in both methodology and problem formulation.
> In particular, **Verisig and POLAR address reachability analysis of neural-network-controlled systems**, computing flowpipes of closed-loop systems in which the neural network acts as a controller. In contrast, **our approach assesses $\epsilon$-closeness of a neural network to a known function** for use as a neural abstraction. The key challenge in assessing $\epsilon$-closeness arises in **verifying across a large domain**, which necessitates refinement and which has *not* been addressed in any neural-network-controlled systems reachability analysis tool.
>
> These problems differ not only in their specifications, but also in the role and treatment of the Taylor model. Namely, **in Verisig 2.0 and POLAR, Taylor models are propagated through the neural network** to approximate the controller’s behaviour, enabling analysis of the composite closed-loop dynamics using tools such as Flow*. By contrast, **our method applies certified first-order Taylor models to relax an analytically defined nonlinear function into a linear surrogate specification** (Theorem 1), which is then verified against a neural approximation.
>
> We further highlight the distinction in specification and methodology below.
>
>  - Our method introduces a refinement strategy that adaptively partitions the input space to reduce conservatism. This refinement, combined with parallel verification, allows our approach to scale to higher-dimensional systems (up to 7D) and handle broader application classes such as neural network compression.
>
>  - In our paper, we draw attention to multiple tools that exist for relaxing the nonlinearity of neural networks, such as Linear Bound Propagation, methods based on Lipschitz constants, and Interval Bound Propagation. We acknowledge that we have not referenced specific tools that use Taylor models for neural networks (e.g., *Verisig*, *ReachNN*, *Verisig 2.0*, *POLAR*). We will include appropriate references to these works. However, we would like to stress that the novelty of **our contribution is independent of the particular neural network verification tool used**.
>
>  - Prior work has already demonstrated how neural abstractions can be composed with Flow* to analyse the behaviour of autonomous dynamics governed by a neural abstraction. We have therefore focused on the verification task of proving the validity of a neural abstraction itself.
>
> To our knowledge, **the $\epsilon$-closeness specification has until now only been treated in Abate et. al. (2022) [1]**, in which **an SMT solver was required** to handle nonlinear real arithmetic theory as the specification involved the nonlinear function $f(x)$.
>
> ---
>
> [1] Alessandro Abate, Alec Edwards, and Mirco Giacobbe. Neural abstractions. In Proceedings of the
> 36th International Conference on Neural Information Processing Systems, NIPS ’22, Red Hook,
> NY, USA, 2022. Curran Associates Inc. ISBN 9781713871088.

---

> > ### Comment · Reviewer_PF2z · 2025-11-26
> >
> > Thank the authors for the response. While the differences in the research problems addressed in this paper and in POLAR/VERISIG are clear, the technical challenge is not fully clarified. I am curious whether these techniques (including Flow*, thanks for reminding me of this) can be easily adapted to address the problem mentioned in this paper. If not, what is the fundamental challenge there? If yes, how is their performance?

---

> ### Author Response · Authors · 2025-11-25
>
> Dear Reviewer, we would be happy to engage in a discussion about the differences of our problem setting and the tools suggested. :-)

---

> ### Author Response · Authors · 2025-11-26
> **Follow-up Response to Reviewer**
>
> We thank the reviewer for the follow-up questions and address them in turn.
>
> ### Fundamental challenge addressed by POLAR/VERISIG vs. our work.
>
> To apply Flow\* to closed-loop systems that use neural network controllers, a polynomial representation of the neural network controller is required. This is the fundamental challenge addressed by Verisig and Polar: finding a local polynomial representation of possible control actions by a neural network controller taken over a small set of states, at each step of the flowpipe computation within Flow\*. To address this, frameworks such as *Verisig* and *POLAR* construct *local* Taylor models of the neural-network controller relying on the key assumption:
>
> >  The input region to each Taylor model is sufficiently small, so that the Taylor remainder remains well-behaved.
>
> By contrast, our problem is fundamentally different:
>
> > We certify uniform $\varepsilon$-closeness between a neural network $N(x)$ and an analytic function $f(x)$ over a \emph{large, global input domain} $X$.
>
> The challenge of this problem is that it requires finding the global optimum of the nonlinear, nonconvex optimization problem: the maximum distance $\max_{x \in X} |f(x) - N(x)|$ is less than $\epsilon$? Over such large domains, Taylor remainders grow quickly unless one introduces *adaptive partitioning* (one of our contributions), which is absent in *Verisig*/*POLAR*. Our refinement is explicitly driven by (i) Taylor conservatism and (ii) neural-network verification outcomes, which are not captured by existing tools. In addition to this global-to-local decomposition via branch-and-bound (our certificate refinement scheme), to solve this global optimization problem, we require integration with NN verifiers such as Marabou, another key contribution.
>
> To briefly summarize: existing TM-based tools solve a *local reachability* problem, not the *global closeness* problem.
>
> ### Could techniques from Verisig/POLAR help?
> Yes, certain techniques, such as shrink-wrapping or symbolic remainder manipulation, could tighten our local linear bounds and potentially decrease the number of required partitions.
> However, once the domain has been sufficiently refined, the dominant computational cost is the NN verification rather than the linearization tightness. Thus, while these tools would provide incremental improvements in reducing the number of partitions,  we do not expect a significant reduction in runtime, as these techniques do not address the fundamental difficulty of the global closeness verification problem.
>
> Does this address the reviewers' remaining concerns? We will update the manuscript to reflect this positioning with respect to the neural network controller literature.

---

### Official Review · Reviewer_GfUf · 2025-10-30

**Soundness:** 4
**Presentation:** 4
**Contribution:** 3
**Rating:** 6
**Confidence:** 4

**Summary:**

The paper proposes techniques for training and certifying neural network approximations of dynamical systems. The proposed method (i) constructs a loss function from the difference between the discrete- or continuous-time dynamics and the NN approximation, (ii) uses a Taylor-series approximation over hyperrectangle partitions and linear satisfiability checkers to check the approximation accuracy, and (iii) refines the hyperrectangle partition as needed to avoid false negatives in verification. The approach is evaluated on a collection of nonlinear ODE test systems to demonstrate scalability.

**Strengths:**

The approach is sound and improves on standard techniques such as dReal; to the best of my knowledge, this approach does not appear elsewhere in the literature and results in substantial speedup by converting nonlinear problems to (relaxed) linear ones. The presentation of the paper is easy to follow. The experiments include some nonlinear systems that scale up to seven dimensions, as well as a neural network compression example.

**Weaknesses:**

I have some questions regarding the results and contributions of the paper that are listed below.

**Questions:**

1. The paper claims to be about approximating dynamical systems, however, it appears to be about the more general problem of certifying the accuracy of a neural network approximation N to a known function f, with dynamics approximation as a motivating application. Can the authors comment on the generality of the results, and how they relate to the literature on function approximation via neural networks?

2. How does the paper compare to related works that use Taylor approximations and interval bounds for neural network reachability analysis (e.g., Verisig)?

3. In order to certify the results, upper and lower bounds on the Taylor approximation error are needed, which the authors compute by bounding the norm of the Hessian. In general, this requires solving a nonlinear optimization problem. How does the cost of computing these Taylor series bounds compare to the cost of directly solving the NN verification problem max{ ||f(x)-N(x)|| : x \in H_\delta(c)}?

4. I would be interested if the authors can comment on some of the simulation results. In particular, the fact that the 7-dimensional spacecraft model takes roughly the same order of magnitude to verify as the 3-dimensional steam governor is somewhat surprising.

---

> ### Author Response · Authors · 2025-11-18
>
> We thank the reviewer for the time and effort dedicated to carefully reviewing our paper.
>
> ## Question 1) Generality of results
> We agree that our verification framework is a general method for certifying the accuracy of function approximations---we view this as a significant strength. In our paper, we decided to focus specifically on dynamical systems due to the importance of the setting, in particular its use in constructing neural abstractions, and of the specific associated challenges. The key challenge in verifying $\epsilon$-closeness is that the state spaces are large, dense input domains with widely varying non-linear function behaviour across the domain.
>
> We also demonstrate our method's specific utility for dynamical systems by leveraging their inherent properties. In Section 4.3, we apply our method to the verification of learned Koopman embeddings, a technique rooted in dynamical systems theory. In this application, we certify the accuracy of entire trajectories---sequences of subsequent states predicted by the model. This task of verifying a "rolled-out" system evolution is unique to the context of dynamical systems.
>
> In the vast body of literature on neural network-based function approximation, most of the literature pertains to the existence of an $\epsilon$-accurate neural network, approximation rates, training, and (empirical) evaluation.
> To the best of our knowledge, this is only the second work after the original Neural Abstractions paper to *formally* quantify $\epsilon$-closeness between a given function and neural network.
>
> ## Question 2) How the paper compares to other works relying on Taylor approximations
> Our approach significantly differs to Verisig both in methodology and in the underlying problem that tools such as Verisig, Verisig 2.0, and ReachNN* handle. These tools consider Reachability Analysis of Neural-Network Controlled Systems, effectively verifying the performance of closed-loop systems containing neural network controllers. Thus, the underlying specification (Closed-Loop Reachability Problem vs Neural Abstraction) is fundamentally different. A key challenge in verifying $\epsilon$-closeness is the requirement of verifying across a large domain, which is not addressed in any of these reachability analysis tools. Furthermore, while Verisig 2.0 employs Taylor Models, the Taylor Model is propagated through the fully connected neural network to obtain a (local) Taylor approximation of the neural network controller to allow the use of Flow* rather than used to relax a non-linear function given in elementary form to obtain a linear surrogate specification (Theorem 1 of our manuscript).
>
> Prior work on neural abstractions has already demonstrated that neural abstractions can be used with Flow* to solve reachability problems of autonomous systems, and as such, we focused on the verification task associated with proving the validity of a neural abstraction, rather than their subsequent usage in control or reachability applications.
>
> ## Question 3) Bounding the norm of Hessians compared to direct optimization
> As highlighted in Appendix D, bounding the norm of the Hessian is merely one method to obtain a first-order Taylor model. In particular, as the regions are refined, our method relies on local information about convexity, concavity, and monotonicity to tighten the Taylor remainder. Furthermore, for many elementary functions, an analytical bound on the norm of the Hessian exists. We will clarify this point in the revision following Proposition 1.
>
> To compare computing Taylor series bounds to the cost of directly solving the verification problem $\sup_{x \in H_\delta(c)} \lVert f(x) - N(x) \rVert$, it is important to point out that the verification problem is a complex non-linear optimization problem, notorious for its computational complexity. Since it is necessary to solve this program to the global optimum, it requires techniques like branch-and-bound and/or interval arithmetic.
>
> ## Question 4) Simulation results and computation time
> That the verification time for the 7D Low Thrust Spacecraft and the 3D Steam Governor are within the same order of magnitude is a testament to the strength of our method: we exploit sparsity patterns, apply adaptive partitioning, and decoupling to accelerate the verification, which is discussed in Section 3.2.
>
> The 3D Steam Governor was chosen as a demonstration of our approach's capabilities of handling highly non-linear and coupled dynamics that require additional refinement to reduce conservatism. On the other hand, the Low Thrust Spacecraft dynamics was chosen to demonstrate that our approach is able to take high order dynamics and leverage there innate linearity in certain dimension and reduce the computation time to that of a lower dimensional nonlinear system. We illustrate this capability in Figure 2, where in the first output requires only a single region for exact representation, as this output is linear or constant in both inputs.

---

### Official Review · Reviewer_mXmt · 2025-11-01

**Soundness:** 3
**Presentation:** 3
**Contribution:** 2
**Rating:** 4
**Confidence:** 2

**Summary:**

This paper addresses the challenge of formally verifying neural networks that serve as approximations (abstractions) of nonlinear dynamical systems. The authors identify that existing state-of-the-art methods relying on Satisfiability Modulo Theories (SMT) solvers over nonlinear real arithmetic (e.g., dReal) face severe computational bottlenecks. To overcome this, the paper proposes a novel framework that replaces global nonlinear reasoning with local, certified linear approximations.

**Strengths:**

1. Significant Scalability Improvements: The shift from nonlinear SMT solving to utilizing established linear verification tools via local approximations is a pragmatic and impactful contribution.
2. Novel Applications for Verification: The paper moves beyond standard control benchmarks to demonstrate versatility.
    - Koopman Operators: Verifying trajectory-level predictions by treating the network as a discrete-time abstraction is a compelling use case.
    - Model Compression: Framing a large "teacher" network as the reference "dynamics" and formally verifying a highly compressed "student" network (98.4% smaller)  is an excellent demonstration of this framework's generality beyond analytical ODEs.
3. Effective Handling of Sparse Nonlinearity: The refinement strategy smartly decouples output dimensions and prioritizes splitting only along relevant input axes. This is highly effective for systems like the Jet Engine benchmark, where nonlinearity is confined to specific components, avoiding unnecessary partitioning of the entire state space.

**Weaknesses:**

1. Dependency on Local Linearity (The "Curse of Partitioning"): While avoiding the complexity of nonlinear SMT, the method trades it for potential combinatorial explosion in partitioning. For highly nonlinear functions where the second-order terms (Hessian) are large everywhere, the hyperrectangles must be extremely small to satisfy the error bounds. The current benchmarks may not fully stress-test this worst-case scenario where every dimension is highly coupled and nonlinear.
2. Koopman Verification Completeness: In the Koopman trajectory experiment, the method certified 99.03% of the domain, finding 29 counterexamples. While identifying safe regions is valuable, for safety-critical applications, the remaining 0.97% is the most important part. The paper does not deeply analyze why verification failed in those specific regions (e.g., genuine failure of the trained model vs. overly conservative bounds that could not be refined further).
3. Compression Benchmark Details: For the compression benchmark, CROWN is used to bound the large teacher network because it is a ReLU network (not twice differentiable). CROWN itself is a relaxation. The paper does not discuss how much additional conservatism (and thus extra partitioning) is introduced by using CROWN for the reference dynamics compared to an analytical function.

**Questions:**

1. The adaptive splitting strategy works well when nonlinearities are sparse or decoupled (e.g., Jet Engine ). Have you analyzed the behavior of your stack-based approach on a "pathological" synthetic system where every output depends highly nonlinearly on every input? At what point does the memory overhead of the stack  become a bottleneck.
2. In the Koopman experiment, 0.97% of the domain remained uncertified. Did you analyze these regions? Are they concentrated around specific state-space features (e.g., boundaries of basins of attraction), or are they scattered due to random approximation errors in the trained network?
3. How did the tightness of CROWN's bounds affect the verification time? Did you find that you needed significantly deeper partitioning for this task compared to benchmarks with analytical derivatives available for Lagrange bounds?

---

> ### Author Response · Authors · 2025-11-18
>
> We thank the reviewer for the constructive comments and for recognizing the scalability gains, versatility, and effectiveness of our proposed verification framework. We take this opportunity to further highlight the conceptual novelty and non-incremental nature of our contributions, which go beyond previous approaches that relied on nonlinear SMT solvers.
>
> ## Question 1) Adaptive splitting on pathological cases and impact of the stack-based approach
>
> The reviewer rightly notes that our approach could, in the worst case, suffer from exponential growth in the number of partitions when faced with highly coupled and nonlinear functions. This exponential blow-up (the curse of dimensionality) is a well-known and fundamental challenge in safety-critical control and formal verification.
> However, we would like to stress that this is precisely what motivates our method's use of the adaptive refinement proposed in Section 3.2., which quickly refines the partitioning such that the Taylor models can exploit local information about convexity, concavity, and monotonicity to achieve significantly tighter bounds compared to Lagrange bounds. This is a key contribution that enables greater scalability compared to state-of-the-art methods, despite the presence of nonlinear dynamics. An example of complex non-linear dynamics with local information is the Exponential System for which $\dot{x}$ is monotone for all $y \leq 0$ and concave for $0.65 \leq y \leq 0.88$.
>
> Regarding the impact of the stack-based approach: the choice of using a stack (as opposed to a queue) corresponds to a depth-first search (DFS) and was made because it is less memory-intensive than a breadth-first search. The memory usage in a depth-first search grows only linearly with the required verification depth (which depends on the system dynamics) and is therefore not expected to be a substantial bottleneck (see Remark at the end of Section 3).
>
> To highlight that our adaptive splitting and stack-based approach work as intended, even with high coupling and nonlinear dynamics, we included the Steam Governor example as it exhibits both properties (see Appendix A.3).
>
> ## Question 2) Koopman verification completeness and analysis of uncertified regions
>
> It is important to note that our usage of certified local linearizations nevertheless allows us to achieve the same level of formal guarantees as prior SMT-based methods rather than relying on heuristics. This reformulation preserves soundness while removing the main computational bottleneck. As such, the counterexamples are indeed *true* counterexamples, i.e., it holds that $\lVert f(x) - N(x) \rVert > \epsilon$, and they are not spurious results of conservative relaxations. This is a key contribution of our approach: in addition to guaranteeing satisfaction over a region, it can prove the existence of a counterexample. In practice, this means that the "uncertified" regions are proven to contain true counterexamples.
>
> To demonstrate the value of our approach in conjunction with partially certified models, we deliberately chose a model for the Koopman verification benchmark with less than 100\% certification. We appreciate the reviewer’s recognition that our approach, unlike previous methods, enables further investigation of regions where verification fails. In response to Question 2, our analysis of the 29 counterexamples for the Koopman benchmark revealed that they reduce to just two points in the input domain, duplicated across multiple output dimensions. These points lie near a corner of the state space---an expectable outcome, as this region was underrepresented during training compared to the centre of the state space.
> Since a detailed analysis was beyond the focus of our paper, we initially omitted this analysis. However, in light of the reviewer’s question, we will include such a discussion in the appendix.
>
> ## Question 3) Impact of conservatism of CROWN bounds
>
> CROWN does not introduce significantly more conservatism relative to certified first-order Taylor models, as both approaches rely on decomposing a complex function into simpler (elementary) functions, building local linearizations, and propagating the bounds through these linearizations. We employ adaptive partitioning to effectively reduce and control the conservatism arising from the linearization. The consequence is that when the known model is a deep neural network, as with the compression benchmark, or a long compositional chain of elementary functions, more regions are needed independently of the specific method for constructing the linear surrogate objective.
>
> As an example of the impact of complex vs simple dynamics with first-order Taylor models, consider the 3D Steam Governor for which verification is only slightly faster than for the 7D Low Thrust Spacecraft, which is a consequence of the Steam Governor's behaviour being more complex and non-linear than that of the Low Thrust Spacecraft.

---

### Author Response · Authors · 2025-12-03
**Final remarks**

We would like to thank the AC for their time and effort in carefully managing and reviewing our submission.
We wish to use this opportunity to emphasize our contributions and express the importance of each component of the proposed method.

## Neural abstractions and global closeness
Neural abstractions is a recent promising framework for building efficient piece-wise affine surrogate models, via ReLU-activated neural networks, of non-linear dynamical systems, Abate et al. (2022). A core challenge in truly scaling this framework is the establishment of a formal *global* bound $\epsilon$ between the non-linear dynamics and the neural network, which previously necessitated the use of an SMT solver with non-linear real arithmetic. In response to the reviewers, we have updated the manuscript (Section 1, page 2, and Section 3 in the revised manuscript) to emphasise
- why previous work on neural abstractions required using SMT solvers, and why this is considered state-of-the-art;
- how our contributions extend the capability beyond the state-of-the-art, thus enabling scalable verification of the global $\epsilon$-closeness property.

Our approach employs certified first-order Taylor expansions, or Taylor models, to *locally* bound the non-linear dynamics with linear approximations. The linear bounds are used to *locally* verify the closeness property using neural network verification techniques, which are subsequently lifted to *global* closeness via an adaptive partitioning scheme.

## Relation to Taylor model reachability analysis
Reviewers PF2z and GfUf asked for a positioning of our work with respect to tools from Taylor model-based reachability analysis, which we have included in the introduction of the revised manuscript. These approaches from the neural network-controlled hybrid system literature differ not only in their specifications but also in the role and treatment of the Taylor model, and we believe this led to an initial misunderstanding of our contributions by reviewer PF2z.

Tools such as POLAR and Verisig perform *local* analysis, requiring that the input region is sufficiently small such that the Taylor remainder remains well-behaved. In our context, since we address a *global* problem, we cannot rely on such an assumption. Instead, we use our adaptive partitioning to intelligently select regions and *ensure* that the Taylor remainder is well-behaved. This distinction has been clarified in the revised manuscript in Section 1, page 2 and we further highlight in Section 3 why SMT solvers were (up until now) considered the state-of-the-art for this problem.

## Generality of results
As recognized by one reviewer, our framework is indeed more general and supports verification of function approximation.
However, we believe that (certified) neural approximations of dynamical systems are an important use case, in particular, for their use in constructing neural abstractions and similar analysis. Furthermore, in this context, the key challenge in verifying $\epsilon$-closeness is that the state space is a large, dense input domain, which necessitates our refinement scheme (Section 3.2).

We also include a benchmark leveraging the inherent properties of the dynamical system (the verification of Koopman embeddings in Section 4.3). In this application, we certify $\epsilon$-closeness for every step in a trajectory, i.e., a sequence of states predicted by the model.

It is also important to stress that the novelty of our framework is independent of the method used to construct first-order models (see Appendix D) and the specific neural network verification tool. Our contribution lies in constructing linear surrogate objectives that can be used with (some) neural network verification tools, as well as in the adaptive partitioning scheme that allows the local guarantees to be lifted to the global domain to solve the $\epsilon$-closeness problem.

---

### Meta-Review · Area_Chair_Nn3U · 2026-01-08

**Summary:**

This paper proposes a verification framework based on certified first-order models for neural approximations of nonlinear dynamical systems. With an adaptive partitioning scheme, the framework enables more scalable verification of global $\epsilon$-closeness property over the SMT-based method in the literature. The reviewers acknowledged the soundness of the proposed approach, but also raised concerns on the dependency on local linearity, the details of the two applications in Koopman operators and model compression, the lack of comparison with the literature (e.g., tools such as Verisig and POLAR), and the limited scope of the experiments.

**Reviewer Concerns:**

The authors clarified some of the technical details and explained the difference between their approach and those based on solving local reachability problem (Verisig, POLAR, etc.). While these should address some of reviewers' questions, their concerns on the lack of empirical comparison with those methods in the literature and more comprehensive experiments where the proposed method can be fully stress-tested would likely remain.

**Reviewer Scores:**

Reviewer GfUf has an initial score of 6 and did not respond. Reviewer RDFo has an initial score of 4. They responded to the rebuttal and kept the score. Reviewer PF2z has an initial score of 2. They responded to the rebuttal and remained skeptical. The authors provided additional explanation afterwards, but those would unlikely change the score to the positive side, given the lack of empirical comparison with those methods based on local reachability analysis and the need of more evidence to show the advantages of solving the global $\epsilon$-closeness property. Reviewer mXmt has an initial score of 4 and did not respond. Given their doubt on the current benchmarks, their willingness to raise the score is perhaps 50/50.

---

### Decision · Program_Chairs · 2026-01-26

Reject